# Prevalence and factors associated with Tungiasis among school age children in Sub Saharan Africa: A systematic review and meta-analysis

Gete Berihun [1]*, Belay Desye[2], Leykun Berhanu [2], Chala Daba [2], Zebader Walle[3], Abebe Kassa Geto [4]

**1** Department of Environmental Health, College of Medicine and Health Sciences, Debre Markos University, Debre Markos, Ethiopia, **2** Department of Environmental Health, College of Medicine and Health Sciences, Wollo University, Dessie, Ethiopia, **3** Department of Public Health, College of Health Sciences, Debre Tabor University, Debre Tabor, Ethiopia, **4** Department of Public Health, College of Health Sciences, Woldia University, Woldia, Ethiopia

* gete_berihun@dmu.edu.et, geteberihun@gmail.com

## Abstract

### Introduction

Tungiasis is an overlooked tropical disease resulting from the penetration of the skin by sand fleas. It leads to significant suffering and can be fatal, particularly affecting school age children and elders, primarily above 60 years old, in rural and urban slums across Sub-Saharan Africa. Despite its great public health consequences, the condition remains largely under reported by the scientific communities mainly in Sub Saharan African countries.

### Objective

To assess the prevalence of Tungiasis and associated factors among school-age children in Sub-Saharan Africa.

### Methods and materials

This systematic review and meta-analysis was done based on the Preferred Reporting Items for Systematic Reviews and Meta-Analysis (PRISMA 2020). Literatures were searched from a variety of databases, including PubMed, Science-Direct, Google Scholar, Hinari, and Google. The eligible studies data were extracted using Microsoft Excel and exported to statistical software, STATA version 14 for further analysis. A random-effect model was considered to estimate the prevalence of Tungiasis. The Egger test and funnel plot were used to evaluate publication bias, whereas $I^2$ statistic was used to measure heterogeneity. The finding of this SRMA was done using 23 selected studies with 9781 study participants.

**Data availability statement:** All relevant data are within the manuscript and its Supporting Information files.

**Funding:** The author(s) received no specific funding for this work.

**Competing interests:** The authors have declared that no competing interests exist.

## Results

This review revealed that the pooled prevalence of Tungiasis was 37.86%% (95% CI: 30.95–44.77; I2 = 98.3%, P < 0.000). In terms of risk factors of Tungiasis, school children who lived with domestic animals (cat or dog) in their home were 2.73 times more likely to affected by Tungiasis compared to those without these pets in their home (OR: 2.73, 95% CI: 1.53–3.94). Additionally, school age children who did not wear shoes at all and wear occasionally were 11.26 (AOR: 11.26, 95% CI: 4.04, 18.49) and 7.61 (OR: 7.61, 95% CI: 3.39, 11.83) more likely to affected by Tungiasis compared to those who were regularly. Finally, school-age children who lived in mud-plastered walls were 4.97 times more likely to be affected by Tungiasis compared to those who lived in cemented wall homes (OR: 4.97, 95% CI: 2.61, 4.61).

## Conclusion

Generally, this systematic review and meta-analysis disclosed that a third of school age children were affected by Tungiasis. Additionally, housing conditions, shoe-wearing practices, and the condition of living with domestic animals were factors significantly associated with Tungiasis. Hence, concerned governmental and non-governmental organizations should work to enhance behavioral modification towards prevention and control of Tungiasis. One-third of the school-aged children were affected by Tungiasis. Contributing factors included inadequate housing conditions, footwear habits, and the presence of domestic animals. Therefore, relevant governmental and non-governmental organizations should promote behavioral changes to prevent and control Tungiasis.

## Introduction

Tungiasis is a neglected disease caused by the penetration of female sand fleas, namely Tunga penetrans and attacking the periungual (nails) region of the human body [1–4].It is a zoonotic disease and endemic, particularly in tropical and subtropical areas, with an estimated of 500 million people at risk of the disease [5,6]. The disease is highly prevalent in rural poor communities and urban slum dwellers due to the availability of various risk factors. It causes severe morbidity and mortality in children aged 5–14 years old, disabled, and the elderly mostly above 60 years old [6–9].

Globally, an estimated of 668 million people are living in areas suitable for Tungiasis and developing countries are the highest victim of the disease [10,11]. The disease is endemic in around 88 countries across the globe and majorities are from African countries. The prevalence of Tungiasis ranges from 60% to 80% among school age children, particularly in low and middle-income countries [5,12–14]. It is highly prevalent in the tropical parts of Africa (Cameroon, Ethiopia, Kenya, Madagascar, Nigeria, and Uganda), Central and South America, and the Caribbean particularly in hot and dry seasons [8,15–17]. More than 75% of the East African population are living in rural areas of highly prone for the disease [7]. The World

Health Organization (WHO) has also included Tungiasis in their NTD roadmap of 2021–2030 [18]. Furthermore, the disease causes for school absenteeism for more than 2.4 million primary school children in Sub Saharan Africa countries [19].

The prevalence of Tungiasis is associated with socio-demographic characteristics, health-seeking behaviors improper waste disposal practices, and lack of community participation in hygiene and sanitation Education [12,20,21]. Additionally, the availability of domestic animal like dogs, cats, pigs, cattle, sheep, horses, mules, rats, and mice near living quarters were taken as considered as factors of Tungiasis. Furthermore housing conditions and behavioral factors were another contributing for the high burden of Tungiasis. Additionally, climatic conditions like prolonged dry seasons are also considered as another risk factors of Tungiasis [3,4,8,9,12,14,21–24].

Tungiasis is characterized by erythema, hyper-erythema, lymphedema, abscesses, ulceration, hypertrophy resulting in desquamation, painful fissures, weakness, loss of toe and toenails, deformation of toes and toenails, and hyperkeratosis [25,26]. Severe forms of the disease may cause fissures, disfigurement, amputation, immobilization, chronic lymphedema, and secondary infections such as septicemia, tetanus, and toxic shock syndrome [7]. Persistent scratching and use of sharp instruments like needles pins, and thorns to extract the fleas may cause for the transmission of blood borne diseases like human immunodeficiency virus (HIV) Hepatitis B and C [7,27]. The complication of the disease may also cause for the multiplication of various bacterial species, including Staphylococcus aurous, Clostridium spp., and Enterobacteriaceae which may produce further consequences for immune-compromised school age children. Additionally, it may also increase risk of getting open wounds which may create anaemia and tetanus, particularly in areas where vaccination coverage is low [28]. The impact also extended to its impairment of quality of life on the families, communities in terms of stigmatization, low school attendance, discomfort, and poverty [6,9,24,29]. The cost of medical treatment further exacerbates its economic burden [27].

Prevention and control of Tungiasis needs a multifaceted approach. Hence, various environmental behavioral measures like strengthening personal and environmental hygiene practices, washing one's feet when wearing closed shoes, proper management of waste, and safe handling of living circumstances daily are relevant strategies to overcome the burden of the disease [30]. Despite various primary studies were conducted in different areas SSA, there is inconsistent findings among different studies. Despite children and elderly are highly for Tungiasis, however less emphasis is given even in endemic areas by the academic communities, healthcare practitioners, decision makers, and donor organizations [3]. Despite its global burden, little attention is given to the disease [24]. To the best of our knowledge, no systematic review and comprehensive study has been carried out on the prevalence and factors associated with Tungiasis in SSA. Therefore, this study aimed to assess the prevalence and factors associated with Tungiasis among school age children in SSA.

### Operational definitions

Tungiasis: based on the Fortaleza classification, Tungiasis is defined as if nodules with black centers, supportive ulcers or punctiform cavities, itching spots, trouble walking, oedema, and skin redness around lesions, loss of toenails or deformed nails were detected during clinical examination [24].

## Methods and materials

### Study setting

The study was conducted in SSA.

### Protocol and registration

The protocol for this systematic review has been registered with the International Prospective Register of Systematic Reviews (PROSPERO), maintained by the University of York Centre for Reviews and Dissemination. The review protocol was registered on December 25, 2023, and is assigned the protocol number **CRD42023493637**.

## Information search and search strategies

This study was conducted based on the Preferred Reporting Items for Systematic Reviews and Meta-Analyses (PRISMA-2020) guidelines [31] (S1 File). A comprehensive search strategy was implemented using several databases, including PubMed/MEDLINE, Science Direct, Hinari, and Google Scholar, to identify pertinent studies. Additionally, a Google search was conducted to locate any extra articles not indexed in these databases. The researchers also examined the reference lists of relevant studies and consulted with subject matter experts to uncover any additional gray literature that might be relevant to the review. For the PubMed/MEDLINE database search, the researchers used a combination of key terms and Boolean operators (AND, OR) to construct a detailed and comprehensive search strategy. (Tungiasis [all field] OR "Jigger infection" [all field] OR "Chigoe flea infestation" [all field] OR "Sand flea infestation" [all field] OR "Chigoe flea disease" [all field] "Chigoe flea bite" [all field] OR "Jigger infestation" [all field] AND ("associated factors" [all field] OR "contributing factors" [all fields] OR "determining factors" [all fields] OR "risk factors" [all fields]) AND ("Angola" [all fields] OR "Benin" [all fields] OR "Botswana" [all fields] OR "Burkina Faso" [all fields]OR "Burundi" [all fields] OR "Cabo Verde" [all fields] OR "Cameroon" [all fields] OR "Central African Republic" [all fields] OR "Chad"[all fields] OR "Comoros"[all fields] OR "Democratic Republic of the Congo" [all fields]OR "Djibouti" [all fields] OR "Equatorial Guinea" [all fields]OR "Eritrea" [all fields] OR "Eswatini" [all fields] OR "Ethiopia" [all fields]OR "Gabon"[all fields]OR "Gambia"[all fields]OR "Ghana"[all fields] OR "Guinea"[all fields] OR"Guinea-Bissau" [all fields] OR "Ivory Coast"[all fields]OR "Kenya" [all fields] OR "Lesotho"[all fields]OR "Liberia" [all fields] OR "Madagascar" [all fields] OR "Malawi" [all fields] OR "Mali"[all fields]OR "Mauritania" [all fields] OR "Mauritius" [all fields] OR "Mozambique"[all fields] OR "Namibia" [all fields]OR "Niger"[all fields]OR "Nigeria"[all fields]OR "Republic of the Congo" [all fields] OR "Rwanda"[all fields]OR "São Tomé and Príncipe"[all fields]OR "Senegal"[all fields]OR "Seychelles" [all fields] OR "Sierra Leone"[all fields]OR "Somalia" [all fields] OR "South Africa"[all fields]OR "South Sudan"[all fields]OR "Sudan"[all fields] OR "Tanzania" [all fields] OR "Togo" [all fields] OR "Uganda"[all fields]OR "Zimbabwe"[all fields]). The search was conducted from March to July 5, 2024.

## Eligibility criteria

**Inclusion criteria:** Articles which fulfilled the following criteria were included in this systematic review and meta-analysis.
   **Population**: The study participants of the study were school-age children.
   **Outcome variables**: The articles must have quantitative data on the prevalence of Tungiasis (yes or no) and factors associated with Tungiasis.
   **Study Design**: The review includes a cross-sectional study design.
   **Study Setting**: The study was conducted in SSA.
   **Language**: Articles which were full text articles published in English language were considered for the study. Only full-text articles published in English were included in the study.
   **Publication Period**: Articles that were published since 1980.

## Exclusion criteria

Qualitative studies, systematic reviews, letters to the editor, short communications, commentaries, and articles not fully accessed after three attempts to contact the corresponding author were excluded from the study.

## Study selection

GB and LB, the two independent reviewers, screened the titles, abstracts, and full texts of the articles to assess the pre-determined set of eligibility criteria. Articles that were deemed eligible by GB & LB were compiled together. However, during a disagreement between these two independent reviewers, a third independent reviewer, BD, was brought in to help make the final determination on whether to include or exclude the article in question.

### Data extraction and management

We employed a standardized data extraction format to collect essential information from eligible studies. The extracted data encompassed the author name, publication year, country, data collection method, sampling technique, sample size, prevalence of Tungiasis, and assessment of the risk of bias. We utilized Endnote reference manager software to organize the search findings and eliminate duplicate articles.

### Quality assessment of studies

The quality of the articles included in the review was evaluated using the Joanna Briggs Institute (JBI) quality assessment tools for analytical cross-sectional studies. The assessment study was conducted based on the following indicators, with response options of yes, no, unclear, and not applicable. (1) inclusion and exclusion criteria; (2) description of the study subject and study setting; (3) use of a valid and reliable method to measure the exposure; (4) standard criteria used for measurement of the condition; (5) identification of confounding factors; (6) development of strategies to deal with confounding factors; (7) use of a valid and reliable method to measure the outcomes; and (8) use of appropriate statistical analysis. The risks for biases were classified as low bias (a score of 6–8), moderate bias (3–5), and high bias (0–2) and articles with moderate and low risks of bias were included in the final review [32] (S2 File).

### Outcome of interest

This systematic review and meta-analysis has review has two outcome variables. The first one is estimation of the pooled prevalence of Tungiasis and the second outcome variable is the risk factors of Tungiasis among school-age children in SSA using a pooled odds ratio (OR) with a 95% confidence interval (CI).

### Handling of missing data

The missing data was addressed through several interventions. First, multiple complete datasets were generated by imputing the missing values several times. In addition, studies with significant amounts of missing data were excluded from the analysis. Moreover, comprehensive data extraction protocols were implemented to reduce the likelihood of missing data during the review process.

### Statistical method and analysis

Data were extracted using a Microsoft Excel spreadsheet and exported to STATA version 14 for further analysis. The heterogeneity of eligible studies was determined using the $I^2$ statistic, with cutoff points 25–50%, 50–75%, and > 75% showing low, moderate, and high heterogeneity, respectively [33]. The pooled estimate of Tungiasis among school-age children was determined using the meta-prop command STATA version 14. Subgroup analysis was conducted using the variables of publication year, country, sample size, geographic location, and study facility (community or school-based). Sensitivity analyses was performed to estimate the influence of each study on the pooled prevalence of Tungiasis among the study participants. Subjective and objective publication bias was determined using the funnel plot test and Egger's regression test, with a p-value <0.05 a 95% CI, respectively [34]. The finding of this systematic review and meta-analysis is presented using graphs, tables, texts, and a forest plot.

## Results

### Searching process

Extensive literature search was conducted across multiple databases and yielded a total of 2,070 studies. After eliminating 876 duplicate articles, 1,194 studies were screened based on their titles and abstracts. Out of these, 1,134 studies were excluded for not meeting the predetermined inclusion criteria. The remaining 60 articles were then evaluated for full-text

eligibility. Following this review, an additional 37 studies were excluded—34 articles did not present the outcome of interest, and 3 were deemed low quality. Ultimately, 23 studies were included in this review (Fig 1).

## Characteristics of studies included in the review

All 23 studies included in the review employed a cross-sectional study design. The total sample size sample size of studies included in the review was 9,781, with each study sample size varies from 143 to 861. The studies were published between 1980 and 2024. All of the studies were done using a similar method of data collection, including face-to-face interviews, observation, and clinical examination. Based on the country, the review included 9 studies were from Kenya [19,23,35–41], 5 studies from Ethiopia [17,24,30,42,43], 3 studies from Nigeria [44–46], 2 studies from Tanzania [5,47], 1 study each from Gabon [48], Rwanda [8], Uganda [49], and Cameroon [50]. The overall description of the studies included in this systematic review and meta-analysis is presented in the table below (Table 1).

## Pooled prevalence of Tungiasis among school age children in SSA

The finding of this review disclosed that the pooled prevalence of Tungiasis among school age children in SSA was 37.86%% (95% CI: 30.95–44.77; $I^2 = 98.3\%$, P < 0.000). This figure was determined using a random effect model and presented visually using a forest plot (Fig 2).

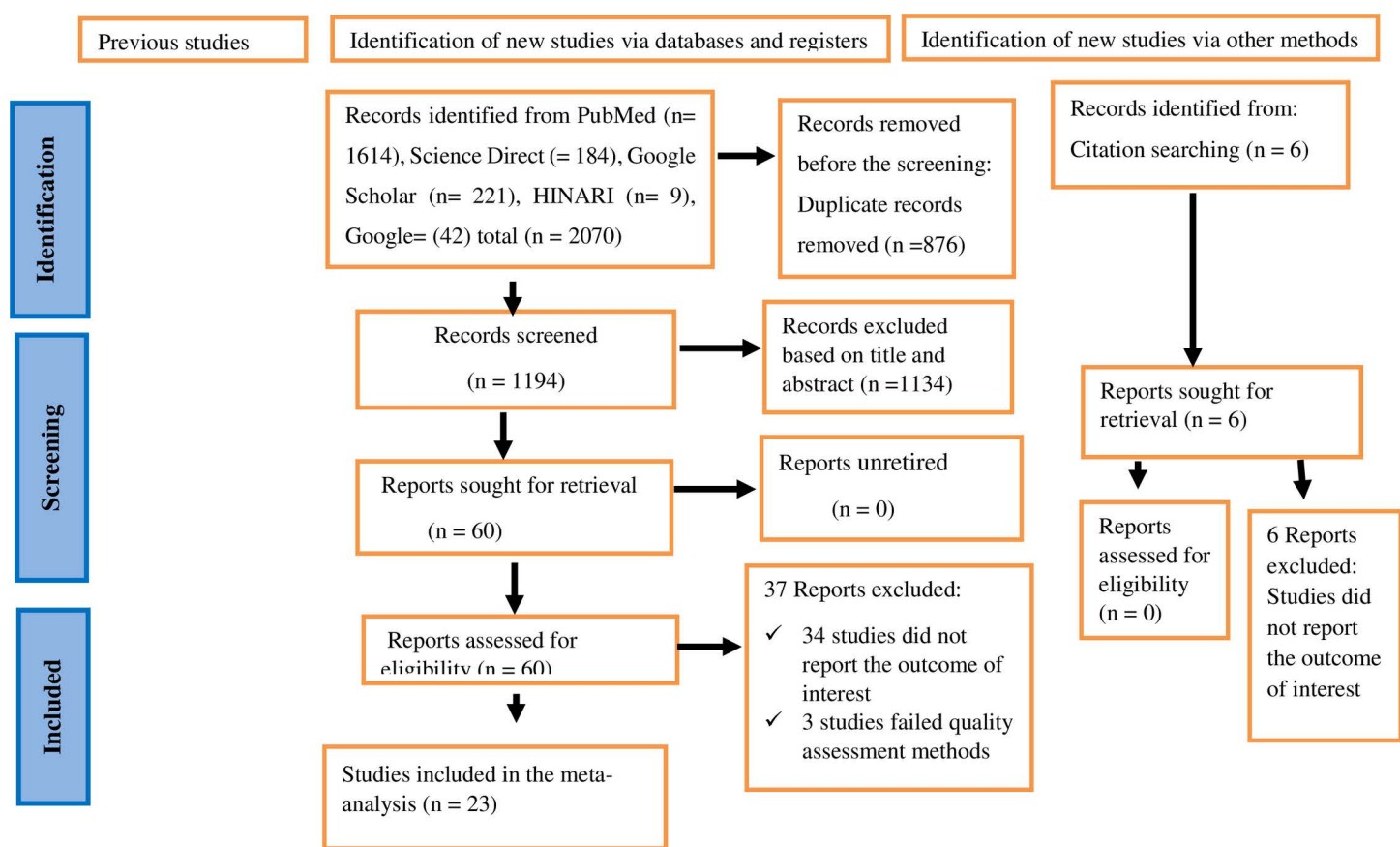

**Fig 1. A PRISMA flow chart showing study selection for systematic review and meta-analysis on the prevalence of Tungiasis and associated factors among school age children in Sub-Saharan Africa.**

**Table 1. Descriptive summary of studies included determination of the prevalence of Tungiasis and associated factors among school age children in SSA in 2024.**

| Author, publication year | Study country | Year cat | Sampling technique | Method of data collection | Study design | Sample size | prevalence | Risk of bias |
|---|---|---|---|---|---|---|---|---|
| Mtunguja (2023) [5] | Tanzania | Since 2020 | multi-stage | clinical examination & structured questionnaire | SBCS | 401 | 21.2 | Low |
| Mwai (2020) [23] | Kenya | Since 2020 | Simple random | semi-structured questionnaire & microscopic examination | SBCS | 384 | 31.1 | Moderate |
| Mwai (2022) [37] | Kenya | Since 2020 | Proportionate stratified random sampling | face-to-face interview microscopic examination | CBCS | 538 | 62.1 | Low |
| Mba (2022) [48] | Gabon | Since 2020 | purposive sampling | face-to-face interview & clinical examination | CBCS | 369 | 36.6 | Low |
| Amare (2021) [42] | Ethiopia | Since 2020 | Simple random | face-to face interview &clinical examination | SBCS | 861 | 54.4 | Low |
| Tamene (2021) [24] | Ethiopia | Since 2020 | systematic random sampling | face-to face interview & clinical examination & observation | SBCS | 487 | 28.3 | Low |
| Jorga (2022) [30] | Ethiopia | Since 2020 | multi-stage | face-to face interview, clinical examination & observation | CBCS | 821 | 52.3 | Low |
| Joseph (2009) [36] | Kenya | before 2020 | prospective cross-sectional | face-to face interview, clinical examination & observation | CBCS | 143 | 21 | Low |
| Walker (2017) [17] | Ethiopia | before 2020 | Undefined | Undefined | SBCS | 343 | 34.7 | Moderate |
| Girma (2018) [43] | Ethiopia | before 2020 | simple random systematic | face-to face interview, clinical examination & observation | CBCS | 366 | 58.7 | Low |
| Nsanzimana (2019) [8] | Rwanda | before 2020 | systematic random sampling | face-to face interview, clinical examination & observation | SBCS | 384 | 23 | Low |
| Ugbomoiko (2017) [45] | Nigeria | before 2020 | Undefined | face-to face interview, clinical examination & observation | SBCS | 545 | 24.4 | Moderate |
| Samuel (2017) [49] | Uganda | before 2020 | simple random | face-to face interview & observation | CBCS | 200 | 12 | Moderate |
| Ngunjiri (2015) [51] | Kenya | before 2020 | simple random &systematic random | face to face interview &clinical examination | CBCS | 347 | 44.1 | Low |
| Mwangi (2015) [15] | Kenya | before 2020 | simple random | face to face interview & examination | SBCS | 508 | 19.1 | Low |
| Ugbomoiko (2007) [52] | Nigeria | before 2020 | simple random | face to face interview & clinical examination | CBCS | 280 | 63.9 | Moderate |
| Mazigo (2012) [47] | Tanzania | before 2020 | simple random | face to face interview &clinical examination | CBCS | 336 | 36.9 | Moderate |
| Anyaele (2021) [44] | Nigeria | Since 2020 | simple random | face to face interview &clinical examination | CBCS | 341 | 50.7 | Moderate |
| Wiese (2017) [41] | Kenya | before 2020 | simple random | face to face interview &clinical examination | CBCS | 425 | 34.8 | Low |
| Okoth (2015) [53] | Kenya | before 2020 | simple random | face-to face interview, clinical examination &observation | CBCS | 143 | 33.6 | Low |
| Albert (2016) [35] | Kenya | before 2020 | purposive sampling | face-to face interview, clinical examination & observation | CBCS | 477 | 41.7 | Moderate |
| Collins (2009) [50] | Cameroon | before 2020 | simple random | face-to face interview, clinical examination & observation | CBCS | 586 | 66.7 | Moderate |
| Ndung'u (2015) [39] | Kenya | before 2020 | purposive and random sampling | face-to face interview, clinical examination & observation | CBCS | 496 | 19.4 | Moderate |

FI = face to face interview, CE = clinical examination, Ob = observation, CBCS = community based cross-sectional, SBCS = school based cross sectional.

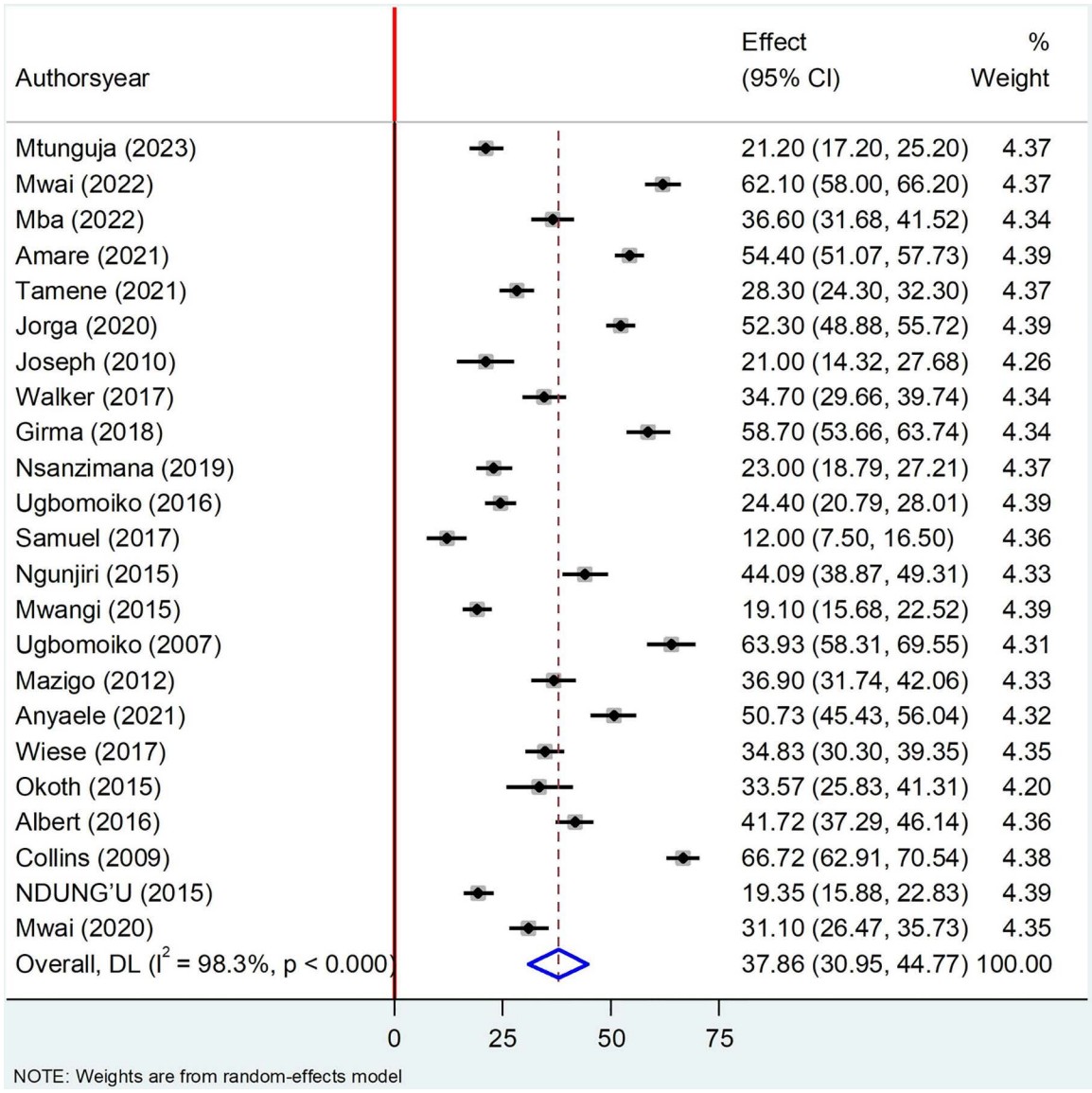

**Fig 2. Forest plot of pooled prevalence of Tungiasis among school age children in SSA in 2024.**

## Sub group analysis

The subgroup analysis is important for identifying potential sources of heterogeneity among the included studies. The subgroup analysis was done using different variables, including country, geographic location (East Africa, West and Central Africa), sample size (below the median and median and above median), study settings (school based or community based), and study year (before and after 2020).

## Subgroup analysis by geographic region

The subgroup analysis based on geographic location revealed that the prevalence of Tungiasis among school age children in East Africa was 34.91% (95%CI: 27.61, 42.21) $I^2 = 98.0\%$, P < 0.000 and West and Central Africa 48.45% (95%CI: 30.76, 66.13) $I^2 = 98.7\%$, P < 0.000 (Fig 3).

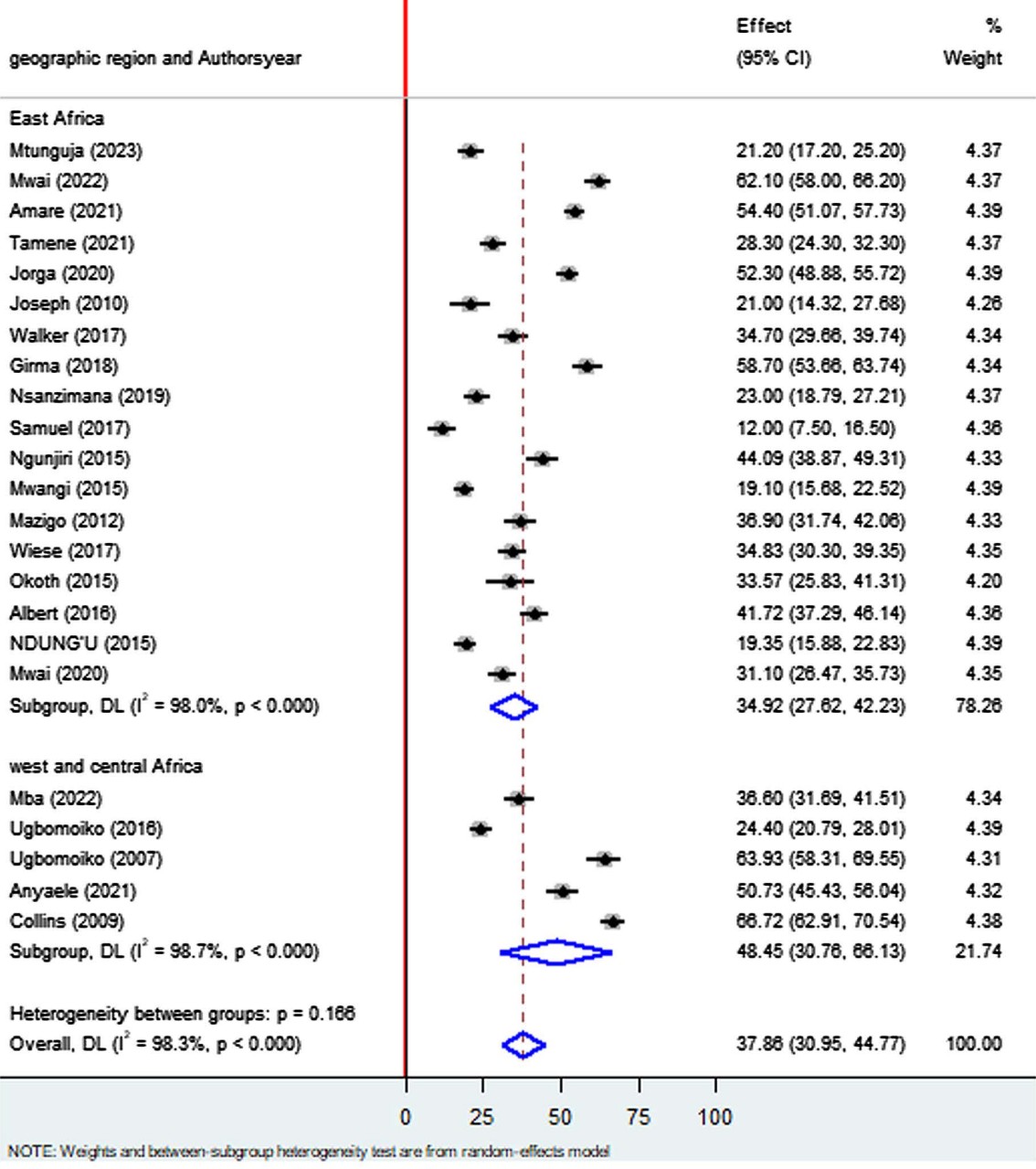

**Fig 3. Forest plot of sub-group analysis based on geographic location on prevalence of Tungiasis and associated factors among school age children in SSA in 2024.**

## Sub-group analysis by country

Regarding the subgroup analysis using country, the prevalence of Tungiasis among school age children in Tanzania was 29.96% (95%CI:13.57%, 44.38) $I^2$ = 95.8%, P < 0.000; Kenya 34.09 (95%CI:23.91, 44.28) $I^2$ = 97.8%, P < 0.000; Ethiopia 45.69% (95%CI: 34.45, 56.93) $I^2$ = 97.4%, P < 0.000; Nigeria 46.28% (95%CI: 21.78, 70.78) $I^2$ 98.7%, P < 0.000, and other

countries 34.57% (95%CI:9.74;59.40) $I^2$ = 99.2%, P<0.000. The subgroup analysis by country showed significant variation in Tungiasis prevalence, ranging from a low of 12.0% in Uganda to a high of 68.72% in Cameroon (Fig 4). **Others (Uganda, Cameroon, Rwanda, and Gabon)**

### Sub group analysis by publication year

The subgroup analysis by study year revealed that the prevalence of Tungiasis in school age children in studies done before 2020 was 35.58% (95%CI: 26.67, 44.49), $I^2$ = 98.3%, P<0.000 and 2020 whereas studies done since 2020 42.08% (95%CI: 28.37, 45.98) $I^2$ 97.1%, P<0.000 (**Fig 5**).

### Subgroup analysis by sample size

The subgroup analysis based on the study sample size, the prevalence of Tungiasis among school age children above the median sample size was 38.75% (95%CI: 27.85, 49.30) $I^2$ = 98.9%, P<0.000 and below the median sample size was 37.17% (95%CI: 28.37, 45.98) $I^2$ = 97.1%, P<0.000 (Fig 6).

### Subgroup analysis by study facilities

Based on the study facilities, the prevalence of Tungiasis among school age children in school-based studies were 29.50% (95%CI: 20.56, 38.44) $I^2$ = 97.6%, P<0.000 and community-based studies showed the prevalence of 42.33% (95%CI: 33.32, 51.34) $I^2$ 98.2%, P<0.000 (Fig 7).

### Sensitivity analysis

The sensitivity analysis showed that no single study had a disproportionate effect on the pooled prevalence of Tungiasis among school age children in SSA. This suggests indicates the pooled estimate is robust and not overly influenced by any individual study included in the analysis which implies that the pooled prevalence remained relatively unchanged when any single study was excluded indicating the overall finding is stable and trustworthy (Fig 8).

### Heterogeneity and publication bias

The existence of heterogeneity and publication bias among studies were determined. This finding revealed studies included in the review showed a high level of heterogeneity ($I^2$ = 98.3%, p<0.000). The finding of the publication bias indicated that the funnel plot appeared reasonably symmetrical, and the Egger regression test did not show a statistically significant result, with a p-value of 0.795 which shows that the absence of publication bias among the included studies in the review (Fig 9).

### Factors associated with school age children Tungiasis in SSA

Based on the finding of this review, the presence of domestic animals (cat and dog) in their home, and lack of regular shoe-wearing practices, and living in mud-plastered walls were factors significantly associated with the prevalence of Tungiasis among school age children in SSA. Eight studies [5,8,24,30,37,38,42,43] were included to assess factors associated with Tungiasis among school-age children in SSA. Three studies [5,24,43] revealed that school children who live with domestic animals (cat or dog) in their house were 2.73 times more likely to be affected by Tungiasis compared to those who did not shared their residence with domestic animals in their home (OR: 2.73, 95% CI: 1.53–3.94). Additionally, three studies [37,42,43] also revealed that school-age children who did not wear shoes were 11.26 times more likely to be affected by Tungiasis compared to those who wear shoes regularly (OR: 11.26, 95% CI: 4.04, 18.49). Furthermore, three studies [8,15,43] revealed that school-age children who wear shoes occasionally were 7.61 times more likely to develop Tungiasis than those who wear regularly (OR: 7.61, 95% CI: 3.39,11.83). Finally, four studies [8,15,24,30] revealed that

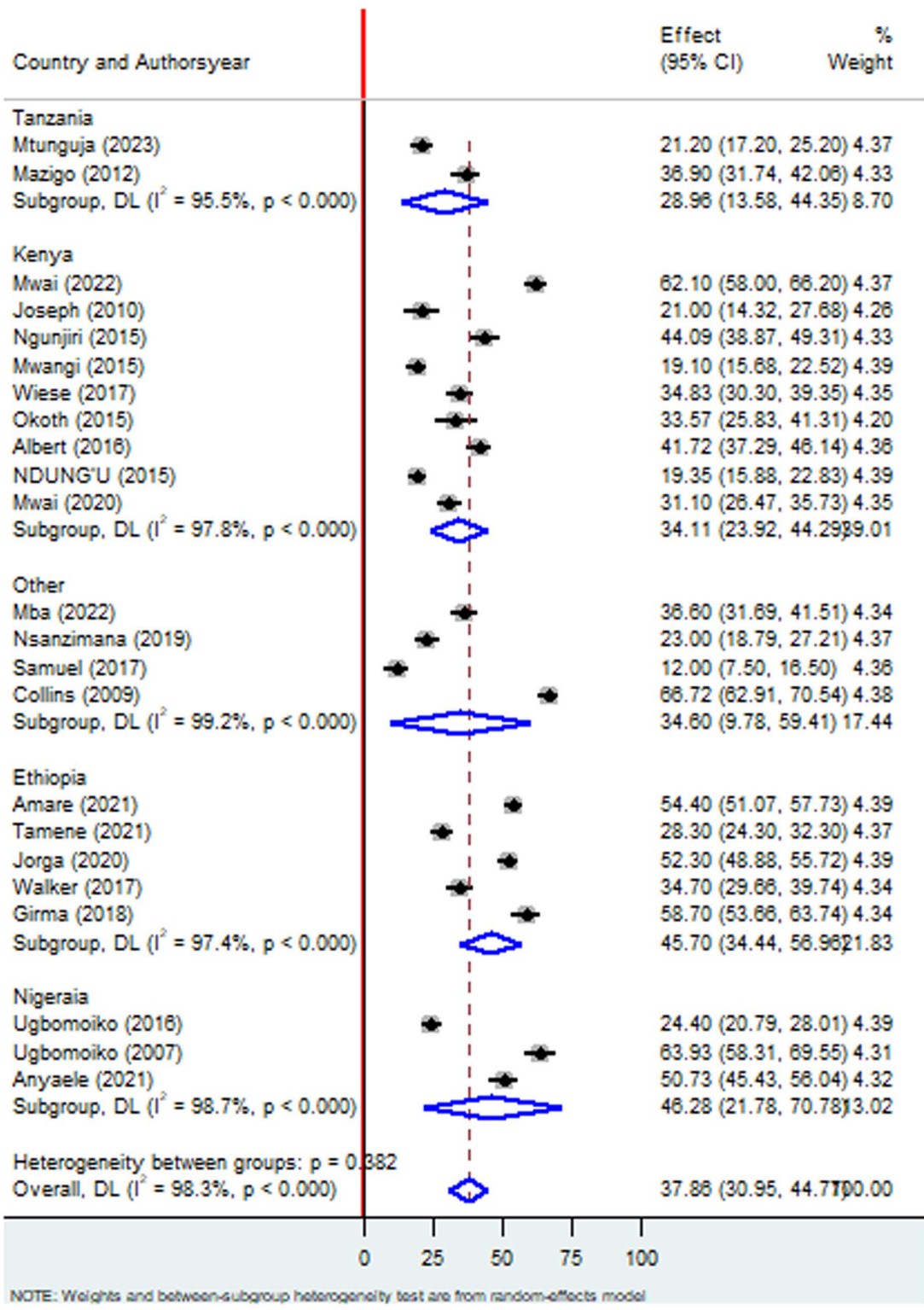

**Fig 4. Forest plot of sub-group analysis based on country on prevalence of Tungiasis and associated factors among school age children in SSA.**

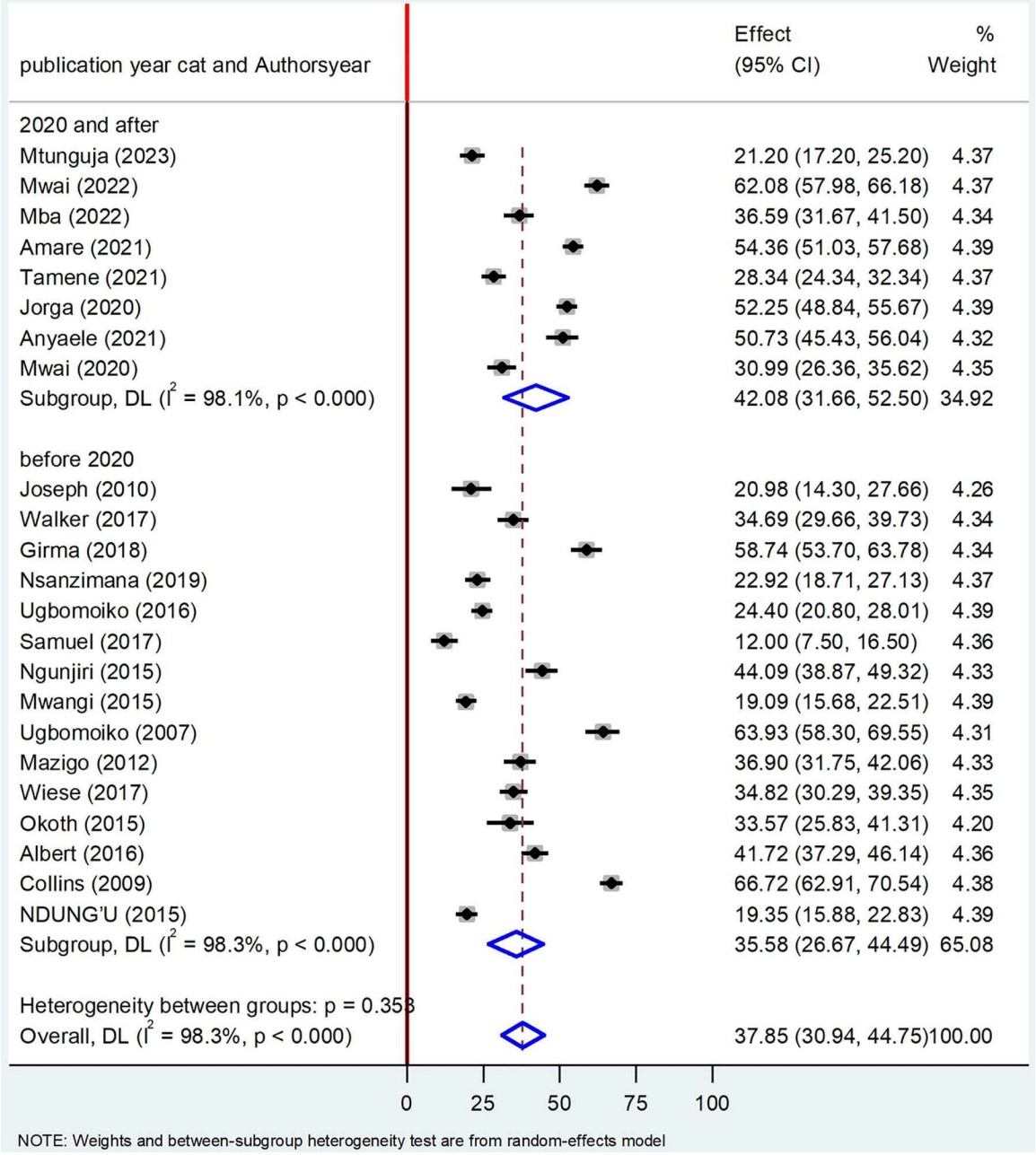

**Fig 5. Forest plot of sub-group analysis based on study year on prevalence of Tungiasis and associated factors among school age children in SSA in 2024.**

school-age children who lived in mud-plastered walls were 4.97 times more likely to be affected by Tungiasis compared to those who lived in cemented walls (OR: 4.97, 95% CI: 2.61,4.61) (Fig 10).

## Discussion

Tungiasis is a neglected tropical disease that is given little attention and recognition by health authorities' health professionals, and other concerned governmental and non-governmental organizations [3,54]. The issue is severe, particularly

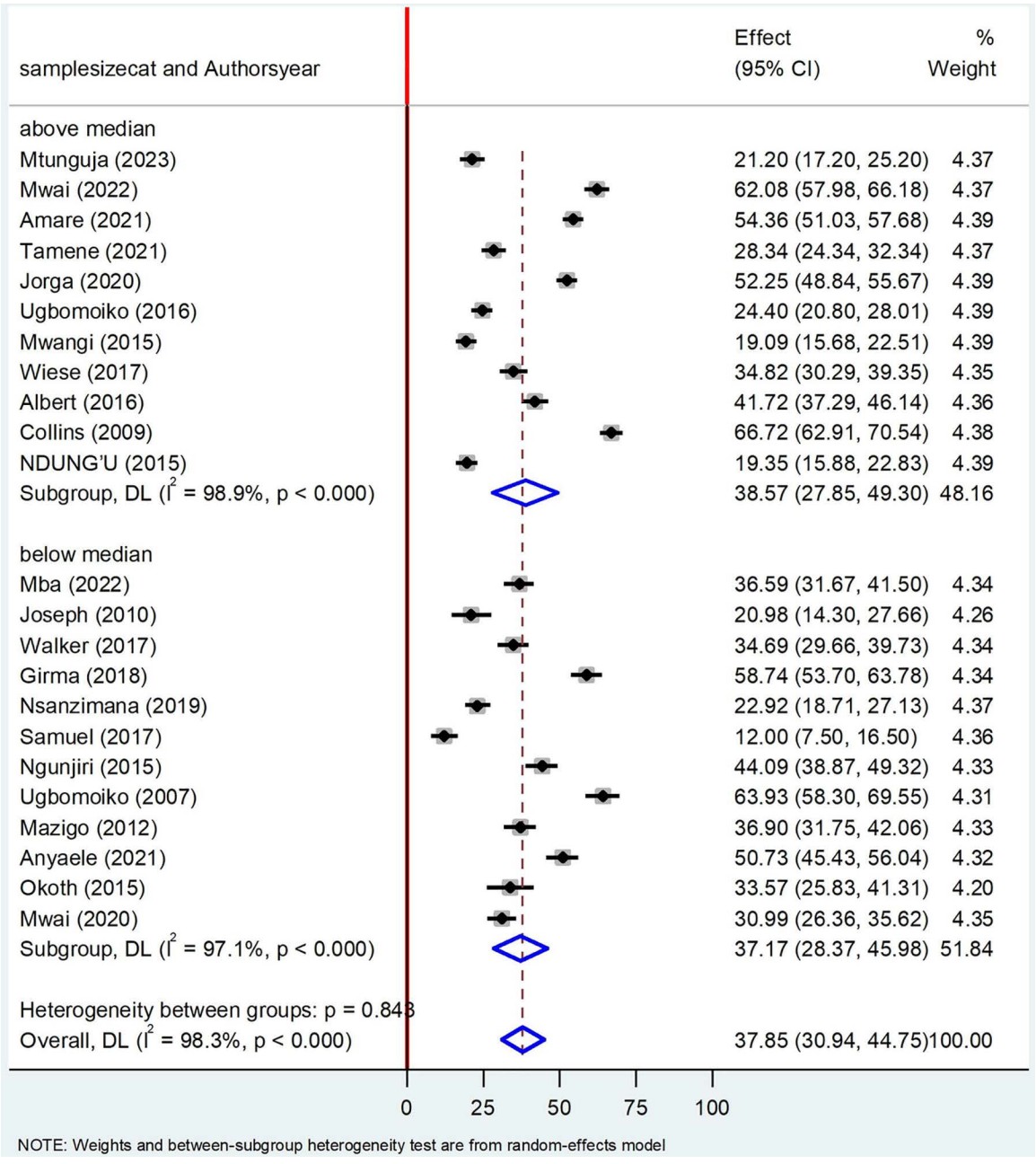

**Fig 6. Forest plot of sub-group analysis based on sample size (below and above the median) on prevalence of Tungiasis and associated factors among school age children in SSA in 2024.**

in resource-limited areas of South America, the Caribbean, and SSA [3,7,12]. Comprehensive prevention and control strategies have not implemented, and there have only been sporadic efforts by non-governmental organizations to treat those affected [5,55]. Hence, this review aimed to assess the prevalence of Tungiasis and associated risk factors among school-age children in SSA.

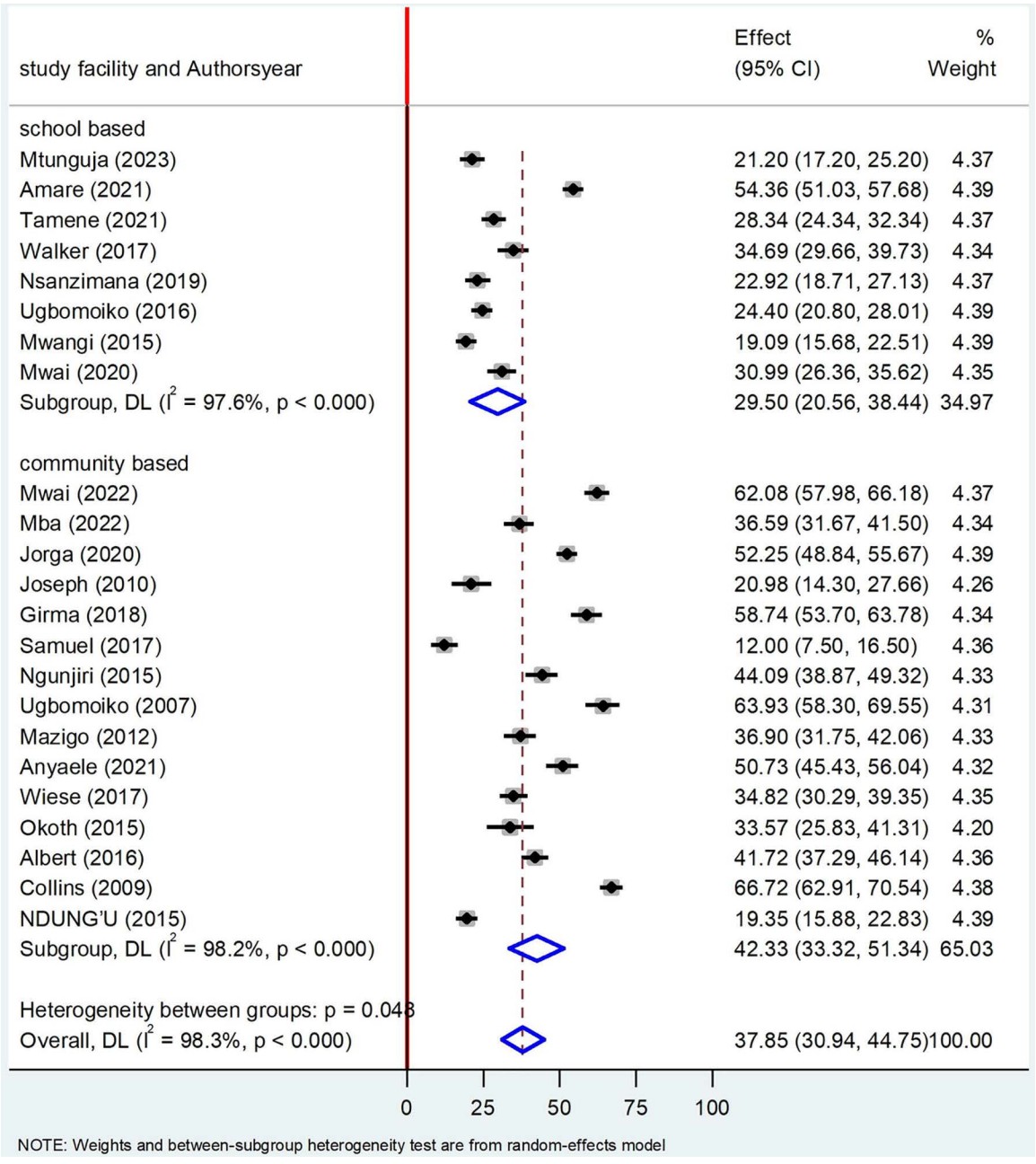

**Fig 7. Forest plot of sub-group analysis based on study facilities on prevalence of Tungiasis and associated factors among school age children in SSA in 2024.**

The finding of this study revealed that the pooled prevalence of Tungiasis among school-age children in SSA was 37.85% (95%CI 30.94, 44.75) which was in line with studies done in Kenya 40% [56], Cameroon (32.7%) [57], in SSA (33.4%) [3]. On the other hand, this finding was higher than studies done in Madagascar (13.7%) [16], Kenya (30.1%) [58], Uganda (28%) [12], (22.5%) [9], (25%) [27], Nigeria (28.4%) [59], Niger delta (30.4%) [60], and Ethiopia (19.5%) [61]. On the contrary, the finding of this study was lower than studies done in Kenya (65%) [62], (85%) [10] and Brazil (51.0%)

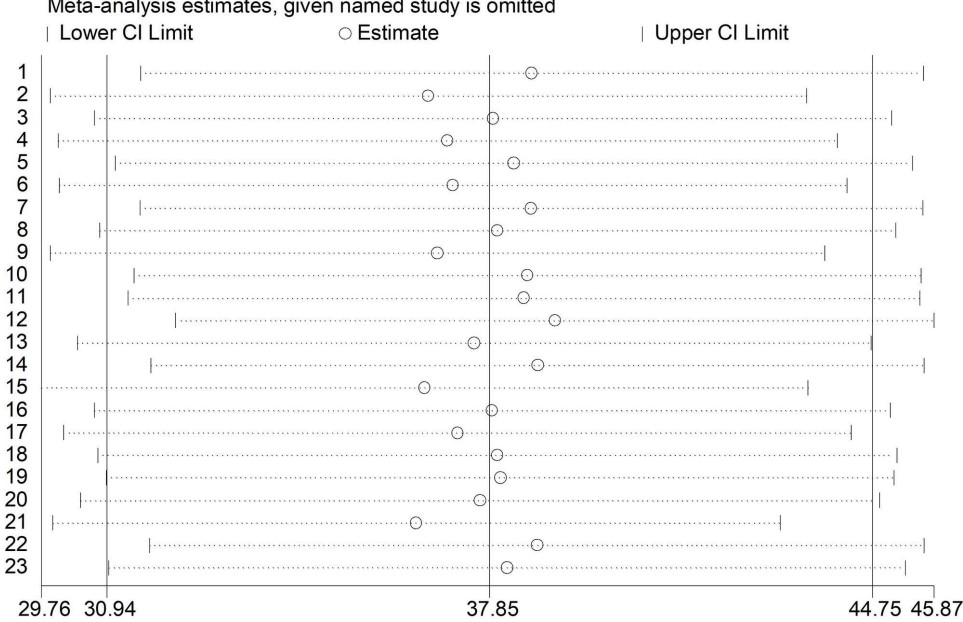

**Fig 8. Forest plot of sensitivity analysis of the pooled prevalence of Tungiasis among school age children in SSA.**

[63], (54.4%) [64]. The possible justification for these variations may be due to the change in exposure characteristics, environmental conditions, and socio-demographic characteristics of the participants [62]. School-age children have limited knowledge about the parasites and usually play in dirty, dusty environments where the parasite often resides. Additionally, the variation in study time and season of the year is considered a factors of Tungiasis [48]. Most children are usually bare-footed, and playful and may have challenges maintaining cleanliness by removing embedded T. penetrans [5,38]. Even if they wore shoes at all, footwear seldom covers the entire foot; open shoes such as sandals and flip-flops, or indeed damaged shoes, are the most common footwear. Children are often left to play in the dry, sandy courtyard, where villagers walk through and spread flea eggs the eggs of the flea [65].

Regarding the subgroup analysis, the prevalence of Tungiasis among school-age children in studies conducted in school setting was 29.50% which was lower than studies done in community based with a prevalence of 42.33%. This deviation may be justified due to the fact that school age children who are infected with Tungiasis may not able to go school. Literatures disclosed that approximately 2.4 million primary school children in SSA are out of school due to differ-ent reasons [51]. The inflammation, pain and itching have been reported to affect children's ability to sleep, walk, attend school and pay attention in class, thus reducing their school performance and social stigma connected with the infestation [24,29]. Regarding study year, studies done before 2020 had a prevalence of 35.58% whereas studies done since 2020 had a high prevalence of Tungiasis 42.08% which implies that the burden of Tungiasis is still increasing and little attention is given by the concerned bodies [24]. These findings support that the disease still is given little attention by the scientific communities which is included in the World Health Organization's (WHO) DNDi Roadmap 2021–2030. Therefore, surveil-lance of its prevalence and comprehensive prevention strategies should be emphasized to divert the progression of the disease across the globe [18]. Regarding country-based prevalence, the highest prevalence was recorded in Cameroon (68.72%), followed by Nigeria (46.28%), and Ethiopia (45.69%), respectively which implies that almost half of the school children were infected with Tungiasis in the above mentioned countries.

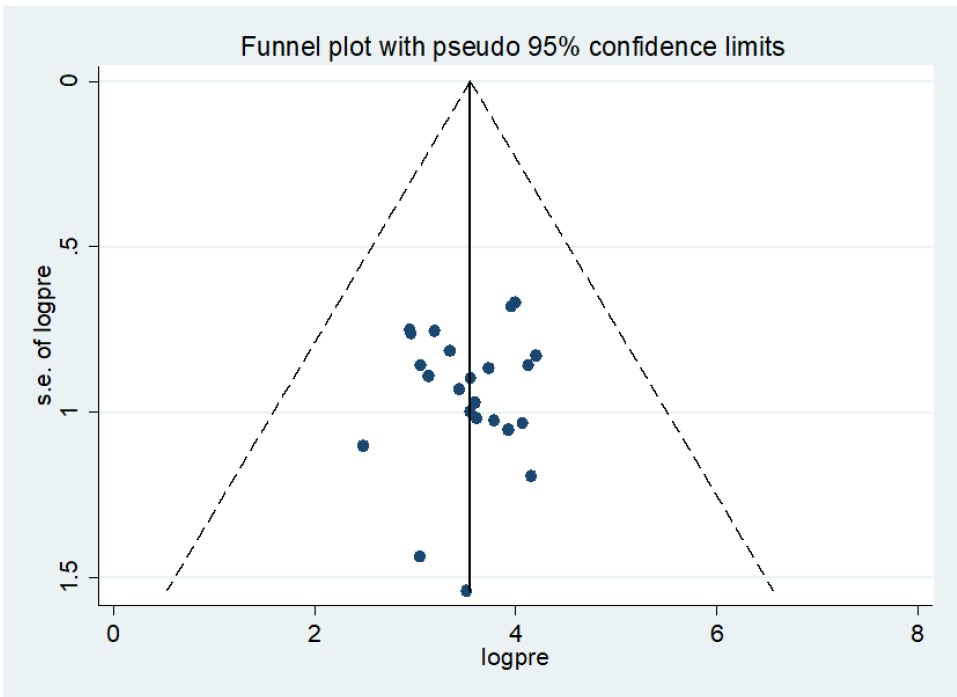

Egger's test for small-study effects:
Regress standard normal deviate of intervention
  effect estimate against its standard error

.
Number of studies = 23                          Root MSE      =   .5233

| Std_Eff | Coef. | Std. Err. | t | P>|t| | [95% Conf. Interval] |
|---|---|---|---|---|---|
| slope | 3.681027 | .5129979 | 7.18 | 0.000 | 2.61419 | 4.747865 |
| bias | -.1503573 | .5719861 | -0.26 | 0.795 | -1.339867 | 1.039153 |

Test of H0: no small-study effects          P = 0.795

**Fig 9. Funnel plot and Egger's regression test, respectively, studies of the pooled prevalence of Tungiasis among school age children in SSA in 2024.**

The finding of this systematic review and meta-analysis revealed that environmental and behavioral factors are declared as factors significantly associated with Tungiasis. Housing condition where school-age children are living (mud-plastered walls), the presence of domestic animals in the household (cats and dogs), and shoe-wearing practices were factors significantly associated with Tungiasis. School-age children living in mud-plastered walled houses were risk factors of Tungiasis which was matched with studies done in Kenya [4,66], Uganda [9], Cameroon [57], and Brazil [65,67]. The trickling of sand and dust from these types of housing conditions creates an ideal environment for the off-host life cycle of sand fleas in cracks in the floor which leads to increased Tungiasis infestation [4]. These types of housing conditions may attract dogs' cats, and other rodents which are an ideal reservoir for sand fleas which are the vectors of

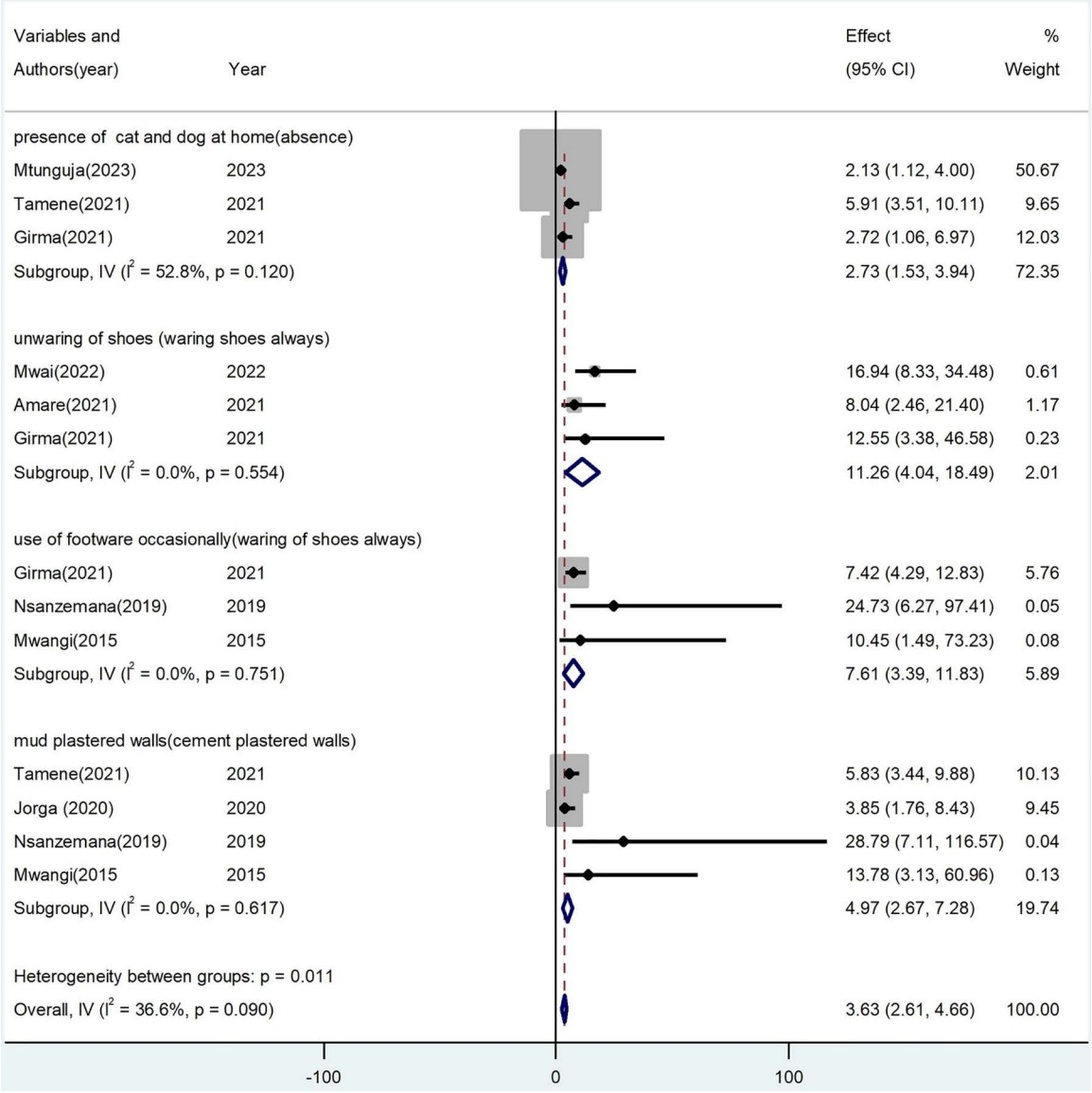

**Fig 10. Forest plot of factors associated with Tungiasis among school age children in SSA in 2024.**

Tungiasis [4,9]. Therefore, the construction of walls made of stone or cement should be taken as a method of prevention and control strategies for Tungiasis [24].

The habit of wearing shoes among school-age children was another risk factor for Tungiasis which was supported by studies done in Kenya [58] and Uganda [9]. Therefore, regular footwear is taken as a strategy for the prevention and control of a range of NTD, including Tungiasis, soil-transmitted helminthes, Mycobacterium ulcerans disease, cutaneous larva migrans, and podoconiosis [29]. The poor habit of wearing shoes among school-age children may be due to the discomfort of walking long distances in closed shoes, especially in rural and hot weather conditions. Additionally, school-age children often prefer to wear open styles of footwear or walk barefoot in many cases, thus long walks to school may be a key factor of Tungiasis [24]. Furthermore, school-age children who were not wearing shoes are exposed to contaminated

soil, predisposing them to Tungiasis, particularly in classrooms and playgrounds, where there is an increased potential for parasite carriers due to high human traffic [37]. Hence, promoting regular wearing of shoes is another important strategy for the prevention and control of Tungiasis infestation.

The finding of this study also revealed that school-age children who lived with domestic animals such as cats and dogs in their house were another risk factor for Tungiasis which was supported by studies done in Brazil [65,67], Kenya [58,68], and Uganda [9]. In many resource-limited countries, human beings usually share residential areas with domestic animals such as pigs, goats, dogs, and cats in their homes. This may be due to inadequate land to shelter animals, fear of theft, and other cultural practices [65]. Different scientific findings indicated that domestic animals are the preferred reservoir for Tungiasis infestation [4,46]. This finding disclosed that Tungiasis is still a global public health concern, particularly in resource limited settings. Hence, there is a need of an urgent need for immediate and long term responses from the global communities to reduce the burden of the problem as much as possible.

### Strength and limitation of the study

The strength of this study stems from its comprehensive literature review, which includes literatures which fulfills the predefined inclusion criteria to overcome the problem of subjectivity in selection of studies. On the contrary, this review has certain limitations, including the use of restricted databases to search for relevant literatures. Additionally, the use of English language written studies may also miss pertinent findings which were published by languages other than English. Furthermore, manuscripts which were under review were also included in the review, which may contain biases in their methodologies, and the conclusions drawn from the research may be revised in the future as it is published.

### Conclusion and recommendation

Generally, more than one-third of school-age children were infested with Tungiasis. The highest prevalence of Tungiasis was reported in Cameron, Nigeria, and Ethiopia. The prevalence of Tungiasis is still increasing as the prevalence of Tungiasis in studies done after 2020 were higher than studies done before. Regarding the factors of Tungiasis, Environmental and behavioral factors were factors of Tungiasis. Sharing of residence with domestic animals like cats and dogs, lack of regular shoe-wearing practices, and living in homes made of mud-plastered walls were factors significantly associated with Tungiasis among school-age children in SSA. Hence, Different prevention strategies focusing on environmental and behavioral modification is should be done by concerned governmental and non-governmental organization should be performed. Awareness creation program should be performed for children and their parents on the over burden and its health impacts. Additionally, cost effective housing conditions program should be implemented and domestic animals should be kept separate from living areas.

### Supporting information

**S1 File. Prisma 2020 checklist.**
(DOCX)

**S2 File. Results of JBI quality assessment.**
(DOCX)

**S3 File. Data set for Tungiasis.**
(XLSX)

**S4 File. Extracted literatures.**
(DOCX)

## Acknowledgments

We would like to express our gratitude to Wollo University for providing the internet access that enabled us to search for and obtain relevant literatures used in this review. We also want to extend our sincere appreciation to our colleagues and friends who provided valuable feedback and insights during the process.

## Author contributions

**Conceptualization:** Gete Berihun, Belay Desye, Leykun Berhanu, Zebader Walle, Abebe Kassa Geto.

**Data curation:** Gete Berihun, Chala Daba, Abebe Kassa Geto.

**Formal analysis:** Gete Berihun.

**Funding acquisition:** Gete Berihun.

**Investigation:** Gete Berihun, Belay Desye, Leykun Berhanu, Zebader Walle, Abebe Kassa Geto.

**Methodology:** Gete Berihun, Belay Desye, Chala Daba, Abebe Kassa Geto.

**Project administration:** Gete Berihun.

**Resources:** Gete Berihun, Leykun Berhanu.

**Software:** Gete Berihun, Belay Desye, Leykun Berhanu, Chala Daba, Zebader Walle, Abebe Kassa Geto.

**Supervision:** Gete Berihun, Belay Desye, Leykun Berhanu, Chala Daba, Zebader Walle, Abebe Kassa Geto.

**Validation:** Gete Berihun, Belay Desye.

**Visualization:** Gete Berihun, Belay Desye, Leykun Berhanu, Chala Daba, Zebader Walle, Abebe Kassa Geto.

**Writing – original draft:** Gete Berihun.

**Writing – review & editing:** Gete Berihun, Belay Desye, Leykun Berhanu, Chala Daba, Zebader Walle, Abebe Kassa Geto.

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
