## [Decision Letter · Decision Letter 0]

29 Oct 2024

PONE-D-24-37631Prevalence and factors associated with Tungiasis among school age children in Sub Saharan Africa: A systematic review and meta-analysis.PLOS ONE

Dear Dr. Berihun,

Thank you for submitting your manuscript to PLOS ONE. After careful consideration, we feel that it has merit but does not fully meet PLOS ONE’s publication criteria as it currently stands. Therefore, we invite you to submit a revised version of the manuscript that addresses the points raised during the review process.

We look forward to receiving your revised manuscript.

Kind regards,

Joshua Kamani, PhD

Academic Editor

PLOS ONE

2. We note that your Data Availability Statement is currently as follows: [All relevant data are within the manuscript and its Supporting Information files.] Please confirm at this time whether or not your submission contains all raw data required to replicate the results of your study. Authors must share the “minimal data set” for their submission. PLOS defines the minimal data set to consist of the data required to replicate all study findings reported in the article, as well as related metadata and methods (https://journals.plos.org/plosone/s/data-availability#loc-minimal-data-set-definition). For example, authors should submit the following data: - The values behind the means, standard deviations and other measures reported; - The values used to build graphs; - The points extracted from images for analysis. Authors do not need to submit their entire data set if only a portion of the data was used in the reported study. If your submission does not contain these data, please either upload them as Supporting Information files or deposit them to a stable, public repository and provide us with the relevant URLs, DOIs, or accession numbers. For a list of recommended repositories, please see https://journals.plos.org/plosone/s/recommended-repositories. If there are ethical or legal restrictions on sharing a de-identified data set, please explain them in detail (e.g., data contain potentially sensitive information, data are owned by a third-party organization, etc.) and who has imposed them (e.g., an ethics committee). Please also provide contact information for a data access committee, ethics committee, or other institutional body to which data requests may be sent. If data are owned by a third party, please indicate how others may request data access. "

3. As required by our policy on Data Availability, please ensure your manuscript or supplementary information includes the following:

Additional Editor Comments:

Dear Authors

Kindly address all the observations raised by the reviewers

Reviewers' comments:

Reviewer's Responses to Questions

**Comments to the Author**

1. Is the manuscript technically sound, and do the data support the conclusions?

Reviewer #1: Yes

2. Has the statistical analysis been performed appropriately and rigorously? 

Reviewer #1: I Don't Know

3. Have the authors made all data underlying the findings in their manuscript fully available?

Reviewer #1: Yes

4. Is the manuscript presented in an intelligible fashion and written in standard English?

Reviewer #1: No

5. Review Comments to the Author

Reviewer #1: The author has done so well in terms of literature review but there is a problem in presenting the information. For example, there is s mix up in the literature right from the introduction. The introduction should talk about the prevalence of Tungiasis and its risk factors. Later it should at least explain what the gap is and how the study is going to deal with the gap

On the methodology, there is repetition on the study setting, the information search is a good one but should be shortened, there is a mix up on the eligibility criteria and main methodology. It would be good if the author uses a diagram in describing the searching process. A good table would also do in explaining the sub group analysis.

There are some grammatical errors in the results and also in the discussion

6. PLOS authors have the option to publish the peer review history of their article (what does this mean? ). If published, this will include your full peer review and any attached files.

**Do you want your identity to be public for this peer review?** For information about this choice, including consent withdrawal, please see our Privacy Policy .

Reviewer #1: **Yes: ** Jarim Oduor Omogi

---

## [Author Response · Author response to Decision Letter 1]

12 Feb 2025

Rebuttal letter

Response to journal requirement

#1. Please ensure that your manuscript meets PLOS ONE's style requirements, including those for file naming.

Response: thank you very much for your concern. Hence, we have tried to revise the manuscript based on the requirement of the journal.

#2. We note that your Data Availability Statement is currently as follows: [All relevant data are within the manuscript and its Supporting Information files.] Please confirm at this time whether or not your submission contains all raw data required to replicate the results of your study. Authors must share the “minimal data set” for their submission.

Response: thank you very much for the confusion we have created. Therefore, we have modified the data availability statement based on the requirement of the PLOS ONE.

#3. As required by our policy on Data Availability, please ensure your manuscript or supplementary information includes the following: A numbered table of all studies identified in the literature search, including those that were excluded from the analyses.

Response: Thank you very much for we have incorporated all the data required for the clarity of the data extraction process.

#4. Please review your reference list to ensure that it is complete and correct. If you have cited papers that have been retracted, please include the rationale for doing so in the manuscript text, or remove these references and replace them with relevant current references. Any changes to the reference list should be mentioned in the rebuttal letter that accompanies your revised manuscript. If you need to cite a retracted article, indicate the article’s retracted status in the References list and also include a citation and full reference for the retraction notice.

Response: We have reviewed all the references and no reference is retracted. Therefore, we have used all the references as it stands and no references are retracted.

Response to reviewer one

#1: The author has done so well in terms of literature review but there is a problem in presenting the information. For example, there is s mix up in the literature right from the introduction. The introduction should talk about the prevalence of Tungiasis and its risk factors. Later it should at least explain what the gap is and how the study is going to deal with the gap

On the methodology, there is repetition on the study setting, the information search is a good one but should be shortened, there is a mix up on the eligibility criteria and main methodology. It would be good if the author uses a diagram in describing the searching process. A good table would also do in explaining the sub group analysis. There are some grammatical errors in the results and also in the discussion.

Response: thank you very much for your constructive comment. We have includes all the comments and concerned raised by the reviewer. See the revised version of the manuscript. Since the aim of this systematic review and meta-analysis, focuses both on prevalence’s and factors associated with the outcome variable of our interest, the introduction section includes all these issues under consideration.

---

## [Editor Report · Decision Letter 1]

7 Mar 2025

Prevalence and factors associated with Tungiasis among school age children in Sub Saharan Africa: A systematic review and meta-analysis.

PONE-D-24-37631R1

Dear Berihun,

We’re pleased to inform you that your manuscript has been judged scientifically suitable for publication and will be formally accepted for publication once it meets all outstanding technical requirements.

Kind regards,

Joshua Kamani, PhD

Academic Editor

PLOS ONE
---

## [Editor Report · Acceptance letter]

PONE-D-24-37631R1

PLOS ONE

Dear Dr. Berihun,

I'm pleased to inform you that your manuscript has been deemed suitable for publication in PLOS ONE. Congratulations! Your manuscript is now being handed over to our production team.

Kind regards,

on behalf of

Dr. Joshua Kamani

Academic Editor

PLOS ONE